Cuproptosis-related lncRNA SNHG16 as a biomarker for the diagnosis and prognosis of head and neck squamous cell carcinoma

Han Baoai 1
Li Shuang 1
Huang Shuo 1
Huang Jing 1
Wu Tingting 1 gracett0310@163.com
Chen Xiong 1 2 zn_chenxiong@126.com
1 Department of Otorhinolaryngology, Head and Neck Surgery, Zhongnan Hospital of Wuhan University , Wuhan , China
2 Sleep Medicine Centre, Zhongnan Hospital of Wuhan University , Wuhan , China
Liu Jinhui
Electronic publication date: 2023 Oct 12
Publication date: 2023
Volume: 11
Electronic Location ID: e16197
Received 2023 Apr 26; Accepted 2023 Sep 6
Copyright: © 2023 Han et al.
Copyright year: 2023
Copyright holder: Han et al.
License: This is an open access article distributed under the terms of the Creative Commons Attribution License, which permits unrestricted use, distribution, reproduction and adaptation in any medium and for any purpose provided that it is properly attributed. For attribution, the original author(s), title, publication source (PeerJ) and either DOI or URL of the article must be cited.
License URL: https://creativecommons.org/licenses/by/4.0/

Keywords: Cuproptosis, SNHG16, Prognostic model, Head and neck squamous cell carcinoma

Funding: Science, Technology and Innovation Seed Fund of Zhongnan Hospital of Wuhan University cxpy2019082 Fundamental Research Funds for the Central Universities 2042023kf0058 Zhongnan Hospital of Wuhan University Excellent Doctor (Postdoctoral) Fund ZNYB2022004 Hubei Provincial Natural Science Foundation 2023AFB408 This research was supported by grants from the Science, Technology and Innovation Seed Fund of Zhongnan Hospital of Wuhan University (No. cxpy2019082), the Fundamental Research Funds for the Central Universities (No. 2042023kf0058), and the Zhongnan Hospital of Wuhan University Excellent Doctor (Postdoctoral) Fund (ZNYB2022004), and the Hubei Provincial Natural Science Foundation (Grant No. 2023AFB408). The funders were significantly involved in the study design, data collection and analysis, decision to publish, and preparation of the manuscript.

==============================
Background

We aim to investigate the potential value of cuproptosis-related lncRNA signaling in predicting clinical prognosis and immunotherapy and its relationship with drug sensitivity in head and neck squamous cell carcinoma (HNSCC).

Methods

We first identified the lncRNAs associated with cuproptosis genes in HNSCC and then conducted a series of analytical studies to investigate the expression and prognostic significance of these lncRNAs. Finally, we used RT-qPCR to validate our findings in a laryngeal squamous cell carcinoma cell line and 12 pairs of laryngeal squamous cell carcinoma and adjacent normal tissues.

Results

We identified 11 differentially expressed lncRNAs that were associated with cuproptosis genes in HNSCC and also served as prognostic markers for this cancer. Enrichment analysis revealed that these lncRNAs were related to immune-related functions that were suppressed in patients with oncogene mutations in the high-risk group. The patients with a high tumor mutation burden exhibited poor overall survival (OS). We used the tumor immune dysfunction and exclusion model to show that the patients in the high-risk group had great potential for immune evasion and less effective immunotherapy. We also identified several drugs that could be effective in treating HNSCC. Experimental validation showed that AC090587.1 and AC012184.3 exhibited differential expression between the TU686 and HBE cell lines, and SNHG16 showed differential expression among the TU686, TU212, and control HBE cells. Among the 12 pairs of cancer and adjacent tissues collected in the clinic, only SNHG16 showed differential expression. Targeted therapy against SNHG16 holds promise as a prospective novel strategy for the clinical management of HNSCC.

Introduction

Head and neck squamous cell carcinoma (HNSCC) is the sixth most common malignancy worldwide, accounting for approximately 890,000 new cases and 450,000 deaths annually (Johnson et al., 2020; Huang et al., 2023a). HNSCC predominantly affects four mucosal surfaces, namely, oral cavity, sinus cavity, pharynx, and larynx, and is strongly associated with smoking, alcohol consumption, and HPV infection. Epstein–Barr virus is closely linked to nasopharyngeal carcinoma (Marur & Forastiere, 2016). Despite significant advancements in the diagnosis and treatment of HNSCC, the lack of early detection and the asymptomatic nature of HNSCC have led to a <50% 5-year survival rate for patients with this cancer. Furthermore, the high risk of recurrence and metastasis of HNSCC persists even after treatment (Bhat, Hyole & Li, 2021; Qin et al., 2021). The advanced stage at which most patients are diagnosed is likely the leading cause of the high mortality rates of HNSCC (Farquhar et al., 2019). The suboptimal preclinical models and absence of biomarkers for early diagnosis severely limit the effective clinical management of HNSCC. Therefore, exploring new diagnostic approaches is imperative to improve the treatment of HNSCC and the quality of life of patients.

Research has recently focused on cuproptosis, a newly discovered form of cell death induced by copper binding to the lipid acylation elements of the tricarboxylic acid (TCA) cycle (Corradi & Mutti, 2011). Copper is an essential catalytic cofactor involved in energy conversion and intracellular oxidative metabolism, and its imbalance can have serious consequences on the development of the heart, central nervous system, and liver metabolism (Ruiz, Libedinsky & Elorza, 2021). Genetic variants that disrupt copper homeostasis may lead to life-threatening diseases, such as Wilson’s disease and Parkinson’s disease (Dubey, Thakur & Chattopadhyay, 2020; Michalczyk & Cymbaluk-Płoska, 2020). Cuproptosis is a unique form of cell death that is distinct from apoptosis and ferroptosis. It is induced by copper binding to the TCA cycle and leads to proteotoxic stress, ultimately resulting in cell death. This discovery sheds new light on the diverse roles of copper in cellular metabolism and the potential implications for human health (Tang, Chen & Kroemer, 2022; Wang, Zhang & Zhou, 2022).

The mechanism of cuproptosis is not yet fully understood. Nevertheless, recent studies have shown that copper can play a significant role in antitumor therapy. In breast cancer, endoplasmic reticulum-targeted copper (II) complexes have been shown to promote the macrophage phagocytosis of cancer cells (Cui et al., 2021). In pancreatic cancer, copper transporter 1 and copper chelator tetrathiomolybdate have been shown to inhibit tumor growth by reducing copper uptake (Yu et al., 2019). These findings suggest that copper has great potential as an antitumor therapy, and exploring the role of cuproptosis in cancer could have enormous clinical application.

LncRNAs are a class of noncoding RNAs that do not encode proteins and are longer than 200 nucleotides. As the largest and most extensively studied class of noncoding RNAs, lncRNAs have become a research hotspot in biomedicine over the past decade (Mercer, Dinger & Mattick, 2009; Kalem & Panepinto, 2022; Peng et al., 2023; Zhang et al., 2023). They exert their biological functions through cell cycle regulation, gene imprinting, mRNA degradation, chromatin remodeling, and splicing and translation regulation. These regulatory processes are associated with tumorigenesis, development, and metastasis, particularly in the early stages of some tumors. LncRNAs can also regulate the expression of cancer-related proteins by mediating gene silencing at the transcriptional level, thereby controlling the invasiveness and apoptosis resistance of tumor cells (Gutschner & Diederichs, 2012; Sanchez Calle et al., 2018; Chi et al., 2019). To date, research on cuproptosis-related lncRNA signatures and their association with OS in patients with HNSCC is limited. To address this gap, our study aimed to construct a prognostic signature of differentially expressed cuproptosis-related lncRNAs based on the TCGA database. We also investigated the immune response of cuproptosis-related lncRNAs and their potential roles in HNSCC prognosis.

Methods

Data preprocessing and filtering

RNA sequencing data, clinical data, and gene mutation data of 502 HNSCC and 44 normal head and neck tissue specimens were obtained from the Genomic Data Commons portal (https://portal.gdc.cancer.gov/) (Jensen et al., 2017). A total of 2,876 lncRNAs in the TCGA cohort were screened according to gene annotation by using 19 cuproptosis-related genes retrieved from previous studies (Table S1). The coexpression correlation between lncRNA and cuproptosis-related genes was analyzed with limma software package was used to analyze the at |R| > 0.4 and P < 0.001 to determine potential cuproptosis-related lncRNAs. A total of 197 cuproptosis-related lncRNAs were identified, and 111 cuproptosis-related lncRNAs with prognostic value were selected through univariate Cox regression analysis on overall survival (OS) with P < 0.05 considered as significant.

Construction of the prognosis model of cuproptosis-related lncRNAs

A prognostic model was developed using cuproptosis-related lncRNA expression data from a training set and tested on another set. LncRNAs’ correlation with survival was significant (P < 0.05) (Li, Bai & Gao, 2023). Eleven crucial lncRNAs were identified using Lasso and Cox regressions (Baliakas et al., 2023). Risk scores determined patient categorization into low or high-risk groups, with Kaplan-Meier curves depicting survival (Bentzen & Vogelius, 2023). ROC and C-index curves validated the model’s accuracy (Yang & Smith, 2023), and a nomogram predicted overall survival (Kang et al., 2023). PCA was constructed to explore the distribution of patients with different risk scores (Khamis & Alanazi, 2023). In addition, the main biological properties of cuproptosis-related lncRNAs were comprehensively examined by Gene Ontology (GO) enrichment and Kyoto Encyclopedia of Genes and Genomes (KEGG) pathway analysis (Gao et al., 2023).

Immunoassay and tumor mutation burden analysis and drug screening

Differences in immune-related function between the high- and low-risk group based on the cuproptosis-related lncRNA characteristics were evaluated using limma and GSVA packages, with P < 0.05 considered as statistically significant. The relationship between high- and low-risk groups and tumor mutation burden (TMB) was compared using Maftools package (Huang et al., 2023b). In addition, the tumor immune dysfunction and exclusion (TIDE) of HNSCC was downloaded from http://tide.dfci.harvard.edu/, and the relationship between the two groups and TMB was analyzed using ggpubr and limma packages.

Verification of screened lncRNAs by qRT-PCR

Human bronchial epithelial (HBE) cell line and laryngeal squamous carcinoma cell lines TU212 and TU686 were obtained from Shanghai Zhong Qiao Xin Zhou Biotechnology. All cells were maintained at 37 °C in a 5% CO2 incubator in RPMI-1640 medium with 10% fetal bovine serum and 1% penicillin-streptomycin. Meanwhile, 12 pairs of fresh tissue samples were obtained from patients who had recently undergone surgical treatment in the Department of Otorhinolaryngology, Head and Neck Surgery, Zhongnan Hospital of Wuhan University from October 2021 to October 2022 (all cases had no other acute and chronic diseases; No prior radiotherapy and/or chemotherapy treatment). The process of collecting fresh tissue samples was as follows: following the basic principles of medical ethics and after obtaining the consent of patients pathologically diagnosed with laryngeal squamous cell carcinoma, the middle and two small tissue samples (about 100 and 50 mg, respectively) were taken from adjacent tissues (5 mm away from the lesion). The blood on the surface was washed with normal saline and transferred to 1.5 ml cryopreservation tubes. The frozen tubes were immediately placed in the liquid nitrogen tank and brought back to the laboratory for storage at 80 °C ultralow temperature refrigerator before being taken out for total RNA extraction. Postoperative pathologically confirmed laryngeal carcinoma tissues and peritumoral tissue specimens were qualified. All the basic and clinical data of the patient were obtained, and T, N, M staging and clinical staging were evaluated according to the patient’s specialist examination, auxiliary examination results, and the “International Union for Anti-Cancer TNM Classification and Staging Standards” (for specific clinical and pathological data, see Table S2). The research protocol was approved by the Ethics Committee of Zhongnan Hospital of Wuhan University (approval number: 2021058), and all patients signed the informed consent. The qPCR primer sequences are listed in Table S3, and β-actin was used as a housekeeping gene.

Results

Screening of cuproptosis-related lncRNAs and construction of a prognostic model

Figure 1 depicts the workflow for establishing the prognostic model and subsequent analysis. Our study comprehensively investigated the pivotal roles and prognostic implications of cuproptosis-related lncRNAs in HNSCC. Initially, we collected mRNA and lncRNA expression profiles and clinical information from 502 HNSCC tissue samples and 44 nontumor samples. After analyzing the expression values of 19 cuproptosis genes and 16,876 lncRNAs, we utilized the rigorous screening criteria of |R| > 0.4 and P < 0.001 and identified 783 cuproptosis-related lncRNAs. We then visually represented the coexpression relationships between cuproptosis genes and cuproptosis-related lncRNAs using a Sankey diagram (Fig. 2A). We integrated the expression data of cuproptosis-related lncRNAs with the corresponding clinical information and then randomly divided the data into training and testing sets. Our aim was to construct a prognostic model for patient outcomes using the training set. As an initial step, we conducted univariate Cox regression analysis to pinpoint the lncRNAs significantly associated with survival. We found 35 lncRNAs that were markedly correlated with survival rates (P < 0.05) (Fig. S1A and Table S4). We employed Lasso regression to further refine our model and mitigate the risk of overfitting, which could introduce false positive parameters. We identified the Lasso coefficients for 20 lncRNAs determined by the optimal lambda value as illustrated in Fig. 2B. The best parameter (lambda) was then visualized with vertical lines in Fig. 2C and Table S5. Multivariate Cox regression was utilized to establish the final model. Our research highlighted 11 lncRNAs as independent prognostic factors for HNSCC. These 11 differentially expressed lncRNAs can act as standalone prognostic predictors for HNSCC (Table S6). Finally, Fig. 2D presents a correlation heatmap, offering supplementary evidence of the association between the mutation-related genes and lncRNAs integrated into the prognostic model.

Figure 1 Flow chart of this study.

Figure 2 Construction of the HNSCC prognostic risk model.

(A) Sankey diagram visualized the coexpression relationships of cuproptosis genes and cuproptosis-related lncRNAs. (B) Lasso coefficient profiles of the 20 lncRNAs with nonzero coefficients determined by the optimal lambda. (C) Optimal parameter (lambda) screening using vertical lines. (D) Correlation heatmap of cuproptosis-related genes and lncRNAs in the prognostic model. Total survival (OS) (E) and progression free survival (PFS) (F) of all patients with HNSCC. (G) Risk score distribution of patients with HNSCC with different risks (low, blue; high, red) in the training group. (H) Dot plots showing the survival time and risk score in the training group. (I) Heatmap of the expression profiles of the 11 key lncRNAs in the training group.

Validation of HNSCC-specific predictive prognostic model

After the prognostic model was established, the patients were divided into high-risk and low-risk groups based on the median risk score. K-M survival analysis was conducted between the training and testing groups and for all groups. The results indicated that OS and PFS were significantly shorter in the high-risk group than in the low-risk group (Figs. 2E and 2F). Moreover, the OS of the high-risk group was lower in the training group (Fig. S1B). To further analyze the survival distribution, we ranked all the patients with HNSCC in the training group according to the risk score (Fig. 2G). A scatter plot illustrated the survival status of patients with different risk scores, with the mortality rate increasing with the risk scores (Fig. 2H). Figure 2I shows the heatmap of the level of 11 lncRNAs in the training group. For example, SNHG16, AC012184.3, LINC01012, AC243965.2, and AP000866.1 are high-risk lncRNAs, and ECE1-AS1, AL358777.3, AL033527.2, AL162458.1, AC090587.1, and CDKN2A-DT are low-risk lncRNAs. We then verified the findings in the testing group, and the results showed that the OS of the high-risk group was significantly shorter than that of the low-risk group (Fig. 3A). In addition, the mortality of the testing group increased with the risk score (Figs. S1C and S1D). The heat map also showed that the expression of cuproptosis-related lncRNAs was associated with the risk score (Fig. 3B). The results for the testing group suggest the reliability of the constructed prognostic model. To determine whether our constructed signal could be used as an independent prognostic factor and is independent of other clinical features, we performed Cox regression analysis on prognostic factors. As shown in Fig. 3C, univariate Cox regression analysis revealed a significant association between OS and age, histological grade, pathological stage, and risk score of patients with HNSCC (P < 0.05). However, upon conducting multiple regression analysis, we observed that only age, N stage, and risk score were the independent prognostic factors significantly associated with OS (P < 0.05) (Fig. 3D). We then employed the ROC curve to assess the predictive accuracy of the risk score, yielding an area under the curve (AUC) of 0.708. The AUC of the risk score was higher than those of age (0.581), gender (0.50), grade (0.544), and stage (0.560) (Fig. 3E). In addition, the AUCs at 1, 3, and 5 years were 0.708, 0.725, and 0.632, respectively (Fig. 3F), signifying that our model was superior in forecasting the prognosis of HNSCC compared with the conventional clinicopathological features.

Figure 3 Validation of the HNSCC prognostic risk model.

(A) Total survival (OS) of patients with HNSCC in the testing group. (B) Heatmap of the expression profiles of the five key genes in the testing group. (C and D) Univariate and multivariate Cox regression analysis of different clinical parameters. (E) ROC curves of different clinical parameters in prognostic model. (F) 1-, 3-, and 5-year ROC curves in prognostic model. (G) C-index curves of different clinical parameters in prognostic model. (H and I) Nomogram for predicting 1-, 3-, and 5-year OS of patients with HNSCC.

Construction of prediction nomogram and principal component analysis

A C-index curve was constructed to compare the consistency index of the risk score with other clinical features such as age, gender, and stage. The results showed that the C-index value of the risk score was higher than those of the other clinical features (Fig. 3G). A nomogram was also developed using age, sex, stage, T, risk score, and N in the signature and accurately predicted the 1-, 3-, and 5-year survival of the patients (Figs. 3H and 3I). The patients in the low-risk group with stages I–II and III–IV exhibited better OS than those in the high-risk group (Figs. 4A and 4B) over time. PCA indicated a high degree of differentiation between the high- and low-risk groups for all genes (Fig. 4C), cuproptosis-related genes (Fig. 4D), cuproptosis-related lncRNAs (Fig. 4E), and cuproptosis-related lncRNA prognostic markers (Fig. 4F).

Figure 4 Principal component analysis and enrichment function analysis.

(A and B) K-M curve showing that the OS of low-risk patients in phase I–II and III–IV was better than that of high-risk patients. Principal component analysis (PCA) of all genes (C), cuproptosis-related genes (D), cuproptosis-related lncRNA (E), and cuproptosis-related lncRNA prognostic markers (F). (G and H) GO and KEGG enrichment analysis of cuproptosis-related lncRNAs. (I) Immune cell correlation analysis of cuproptosis-related lncRNAs.

Functional enrichment analysis and immune-related function analysis

Enrichment analysis revealed significant GO and KEGG pathways for cuproptosis-related lncRNAs. BP analysis showed enrichment in immunoglobulin production, B cell activation regulation, and humoral immune response. MF analysis showed enrichment in antigen binding, immunoglobulin receptor binding, and immunoglobulin binding. Cellular component analysis showed enrichment in immunoglobulin complex, T cell receptor complex, and blood microparticles. KEGG analysis showed enrichment in cytokine–cytokine receptor interaction, cell adhesion molecules, B cell receptor signaling pathway, and NF-kappa B signaling pathway. Furthermore, immune cell correlation analysis showed significant differences in immune cell infiltration between high-risk and low-risk groups. Figures 4G–4I indicate that the cuproptosis-related lncRNAs are associated with immune cell infiltration in HNSCC.

Tumor mutation burden and drug sensitivity analysis

Figures 5A and 5B shows that the mutation frequency was generally higher in the high-risk group than in the low-risk group (e.g., TP53: high-risk, 76%; low-risk, 57%. FAT1: high-risk, 24%; low-risk, 16%. CSMD3: high-risk, 22%; low-risk, 16%). We also found that the TMB was significantly higher in the high-risk group than in the low-risk group (Fig. 5C). TIDE algorithm revealed higher immunotherapy resistance in the high-risk group than in the low-risk group (Fig. 5D). Figure 5E shows that high TMB was associated with low OS in patients. A combined survival analysis of TMB and risk score further demonstrated their significant impact on the OS in patients with HNSCC (Fig. 5F). Finally, potential antitumor drugs were screened using the pRRophetic software package. Analysis of the relationship between risk score and drug IC50 in HNSCC treatment revealed eight drugs with significant sensitivity differences between the high- and low-risk groups. Figures 6A–6C indicate that three drugs (PD-0332991, cytarabine, and WH-4-023) had low IC50s in the high-risk group, implying high sensitivity to drug treatment. These drugs may have a potential role in the future treatment of HNSCC. However, the high IC50 values of other drugs in the high-risk group suggested that the low-risk group was more sensitive to these drugs (Figs. 6D–6H).

Figure 5 Tumor mutation burden analysis.

(A and B) Waterfall diagram showing the mutation in the high- and low-risk group. (C and D) Violin diagram showing that the TMB and TIDE of high-risk group were higher than those of the low-risk group. (E) K-M analysis showing that the OS of patients with high TMB was significantly lower than that of patients with low TMB. (F) Combined survival analysis of TMB and risk score.

Figure 6 Drug sensitivity analysis.

(A–C) PD-0332991, cytarabine, and WH-4-023 had lower IC50 in high-risk HNSCC group. (D–H) YM201636, PIK-93, KIN001-102, phenormin, and THZ-2-495 had higher IC50 in the high-risk HNSCC group.

Validation of lncRNA expression features

After obtaining 11 lncRNAs from the prognostic model, we evaluated their prognostic value. K-M analysis in Fig. 7 indicated that ECE1-AS1, AL033527.2, AL162458.1, and AC090587.1 were positively correlated with OS, suggesting a good prognosis. Meanwhile, SNHG16 and AC012184.3 were negatively correlated with OS, indicating a poor prognosis. Overall, the results of K-M analysis were consistent with those of univariate Cox analysis, indicating that most lncRNAs had strong predictive ability when injected into specific prognostic models.

Figure 7 Kaplan-Meier analyses of cuproptosis-related lncRNAs in prognostic model.

Kaplan-Meier analyses of (A) ECE1-AS1, (B) AL358777.3, (C) AL033527.2, (D) AL162458.1, (E) AC090587.1, (F) CDKN2A-DT, (G) SNHG16, (H) AC012184.3, (I) LINC01012, (J) AC243965.2, and (K) AP000866.1. Statistical significance was determined by log-rank test.

For experimental validation, we ultimately screened three prognostic lncRNAs with consistent expression. First, RT-qPCR was used to verify their expression in cell lines. Figures 8A–8C show that compared with those in the normal HBE line, AC090587.1 was downregulated in the HNSCC cell line TU686. AC012184.3 was upregulated in TU686, and SNHG16 expression was higher in TU686 and TU212 cell lines than in normal cell lines. Among the 12 pairs of clinically collected cancer and adjacent tissues, only SNHG16 showed differential expression (Figs. 8D–8F). These results suggest that SNHG16 can serve as a biomarker for the diagnosis and prognosis of HNSCC.

Figure 8 Verification of the expression of lncRNA in HNSCC cell lines and human tissue by RT-qPCR.

(A–C) AC090587.1, AC012184.3, and SNHG16 expression in cell line; (D–F) AC090587.1, AC012184.3, and SNHG16 expression in human tissues.

Discussion

Conventional treatments can manage this prevalent malignant tumor to enhance patients’ survival time and quality of life. However, within 3–5 years, most patients experience rapid progression with local recurrence and distant metastasis, presenting a risk as high as 30% to 60% and a low survival rate post recurrence and metastasis (Strojan et al., 2015; Wise-Draper et al., 2022). Early identification of high-risk groups for recurrence and metastasis and active clinical intervention can significantly improve patients’ prognosis. In this regard, cuproptosis, a potential therapeutic strategy linked to genetic disorders and tumors, has been implicated in this disease (Tang, Chen & Kroemer, 2022). LncRNAs play a role in various cancers, and different lncRNA predictive signatures can be used to predict the prognosis of patients with cancer (Gutschner & Diederichs, 2012; Sanchez Calle et al., 2018). The function of cuproptosis-related lncRNAs in patients with HNSCC has not been well documented. Our study identified several cuproptosis-related lncRNAs, constructed a reliable prognostic model, and underlined their role in forecasting the prognosis and immune status of patients with HNSCC. We initially identified 19 genes closely linked to cuproptosis and then constructed a prognostic model by analyzing 11 prognostic lncRNAs through a series of biological analyses. ECE1-AS1, AL358777.3, AL033527.2, AL162458.1, AC090587.1, and CDKN2A-DT were identified as protective factors, and SNHG16, AC012184.3, LINC01012, AC243965.2, and AP000866.1 were deemed as risk factors. Certain lncRNAs such as ECE1-AS1, AL358777.3, AL162458.1, AC090587.1, CDKN2A-DT, SNHG16, AC012184.3, and AC243965.2 were identified for the first time. The biological functions of some of these lncRNAs have been confirmed, such as AL033527.2 being linked to immunotherapy response in patients with STAD (Zeng et al., 2022). LINC01012 expression is elevated in gastric cancer tissues, and the proliferation of gastric cancer cells decreased following transfection with siRNA interfering with LINC01012. This finding suggested that LINC01012 may serve as a diagnostic and drug target gene for gastric cancer, providing a new avenue for clinical diagnosis and treatment. Small nucleolar RNA host gene 16 (SNHG16), another lncRNA, is expressed by the SNHG16 gene located on chromosome 17. It was first identified in human neuroblastoma tissue where it was found to promote malignant development (Bao et al., 2020; Ge et al., 2020). In recent years, the pivotal role of SNHG16 in cancer has been gradually uncovered. SNHG16 is a crucial lncRNA that drives the malignant development of various cancers. It is significantly up-regulated in various tumor tissues and cell lines and can promote the development of liver cancer (Li et al., 2021), lung cancer (Chen et al., 2020), and pancreatic cancer (Guo et al., 2020) by mediating the proliferation, migration, and invasion of cancer cells. Our study further confirmed the significance of SNHG16 due to its higher expression in TU686 and TU212 cell lines relative to that in normal cell lines and the significant difference in SNHG16 expression among the 12 pairs of clinically collected cancer and adjacent tissues. This finding reinforced the idea that SNHG16 could become an essential marker and therapeutic target for head and neck tumors, providing new opportunities for precision treatment.

Evidence clarifying the relationship between lncRNAs and HNSCC is insufficient. Our aim was to explore the role of these lncRNAs in HNSCC and their prognostic value. A prognostic model was constructed with 11 lncRNAs, and the patients were categorized into high- and low-risk groups. The high-risk patients had a worse prognosis than the low-risk patients. The reliability of the model was validated through K-M and ROC curve analyses and correlation analysis of clinical variables and risk scores. The nomograms showed that the model was a robust predictor of survival outcomes in patients with HNSCC. Furthermore, we conducted functional enrichment analysis to explore the biological mechanisms underlying the association of the 11 lncRNAs with cuproptosis. KEGG analysis revealed the significant enrichment of these lncRNAs in cytokine–cytokine receptor interaction and NF-kappa B signaling pathways.

Cytokines, including interleukins, interferons, tumor necrosis factor superfamily members, chemokines, and growth factors, are potent secretory regulators involved in cell–cell communication during homeostasis and disease (Borish & Steinke, 2003). They play a crucial role in cellular communication in the tumor microenvironment (Fabris et al., 2021). Although the mechanisms of malignant transformation of various cell types may differ, a plethora of studies have confirmed that the imbalance of the cytokine network resulting from abnormal levels of some cytokines and their receptors is involved in the development and progression of head and neck tumors (Guan et al., 2021; Nisar et al., 2021; Swati & Sharma, 2022). Cytokines can have dual effects on tumors, promoting tumor growth and tumor cell death (Peng et al., 2022; Toma et al., 2022). However, abnormal cytokine expression may contribute to tumor development. When overexpressed, certain cytokines such as IL-1, IL-6, CSF, and EGF can promote uncontrolled cell proliferation and malignant transformation. Alterations in cytokine receptor quantity and signaling pathways can also play a role in tumor formation (Cui et al., 2014; Hirano, 2021; Llovet et al., 2021). The development of head and neck tumors is a complex pathological process that involves multiple factors and steps, including physical and chemical factors, biological factors, genetic factors, and immune status (Baliga et al., 2017; Duprez et al., 2017; Mäkitie et al., 2019). Loss of control of intracellular and extracellular regulatory factors and intracellular information network systems can lead to the disruption of cellular molecules, such as proteins and nucleic acids, hindering the interpretation of cellular language and biological compressed information and thereby causing malignant transformation and the subsequent development of head and neck tumors.

The NF-κB pathway is a receptor pathway regulated by proteolytic enzymes. Numerous studies have demonstrated its close association with tumor development, proliferation, differentiation, apoptosis, invasion, and metastasis (DiDonato, Mercurio & Karin, 2012; Hoesel & Schmid, 2013).

NF-κB activation has been observed in a variety of tumors, such as liver cancer, T-lymphocytic leukemia, breast cancer, cervical squamous cell carcinoma, ovarian cancer, thyroid cancer, Hodgkin’s lymphoma, small cell lung cancer, pancreatic cancer, and malignant melanoma (Runde et al., 2022). Moreover, NF-κB activation is significantly correlated with the development, invasion, and angiogenesis of head and neck tumors. Inhibiting the NF-κB pathway can block the cell cycle and induce apoptosis, thus playing a key role in tumor proliferation (Qin et al., 2018). Immunotherapy is a promising approach for treating head and neck tumors. To guide this approach, we compared the expression of immune checkpoints between high- and low-risk groups and observed significant differences in immune cells, immune checkpoint genes, and immune-related pathways between these groups. These results suggested that the high-risk group has reduced immune function, potentially leading to a poor prognosis and indicating the potential of immunotherapy. We found that that high mutational burden and high-risk scores were associated with a poor prognosis. We also observed that mutant TP53 expression promoted malignant transformation in the high-risk group (Oren & Rotter, 2010; Duffy, Synnott & Crown, 2017). Our findings suggested that high-risk patients may benefit from certain drugs such as PD-0332991, cytarabine, and WH-4-023 while showing resistance to others such as YM201636, PIK-93, KIN001-102, phenomin, and THZ-2-495. These findings lay the groundwork for the precise and individualized treatment of head and neck tumors.

Conclusions

Our study provides novel insights into the pathogenesis and progression of head and neck tumors, with focus on copper homeostasis. We identified cuproptosis biomarkers that could act as prognostic indicators for head and neck tumors and potentially assist in clinical decision-making. However, our prognostic model has some limitations. First, the model needs further validation and improvement with additional data. Second, cuproptosis is a recently discovered cell death mechanism, and research on related lncRNAs is limited. Further studies are needed to elucidate the mechanisms on how the identified lncRNAs regulate cuproptosis.

Supplemental Information

Supplemental Information 1 Supplemental Tables.

Click here for additional data file.

Supplemental Information 2 Supplementary Figure.

(A) Univariate Cox regression analysis was used to analyze cuproptosis-related lncRNAs that were significantly correlated with survival. (B) Total survival (OS) of patients with HNSCC in the training group. (C) Risk score distribution of patients with HNSCC with different risks (low, blue; high, red) in the testing group. (D) Dot plots showing the survival time and risk score in the testing group.

Click here for additional data file.

Supplemental Information 3 Data and code.

Click here for additional data file.

Abbreviations

HNSCC Head and neck squamous cell carcinoma

TMB Tumor mutation burden

TIDE Tumor Immune Dysfunction and Exclusion

OS Overall survival

PFS Progression free survival

ROC Receiver operating characteristic

LASSO Least absolute shrinkage and selection operator

TMB Tumor mutation burden

CNV Copy number variations

Additional Information and Declarations

Competing Interests

Author Contributions

Human Ethics

Data Availability

The authors declare that they have no competing interests.

Baoai Han conceived and designed the experiments, performed the experiments, analyzed the data, authored or reviewed drafts of the article, and approved the final draft.

Shuang Li conceived and designed the experiments, analyzed the data, authored or reviewed drafts of the article, and approved the final draft.

Shuo Huang conceived and designed the experiments, authored or reviewed drafts of the article, and approved the final draft.

Jing Huang analyzed the data, prepared figures and/or tables, and approved the final draft.

Tingting Wu analyzed the data, authored or reviewed drafts of the article, and approved the final draft.

Xiong Chen analyzed the data, authored or reviewed drafts of the article, and approved the final draft.

The following information was supplied relating to ethical approvals (i.e., approving body and any reference numbers):

The research protocol was approved by the Ethics Committee of Zhongnan Hospital of Wuhan University (approval number: 2021058), and all patients signed the informed consent.

The following information was supplied regarding data availability:

The data is available at the TCGA Genomic Data Commons Portal: (https://portal.gdc.cancer.gov/): The HNSCC-related RNA sequencing data, clinical data, and gene mutation data are available under the “Cases by Major Primary Site”, and under “Head and Neck”.

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
