# Peer review of "Cuproptosis-related lncRNA SNHG16 as a biomarker for the diagnosis and prognosis of head and neck squamous cell carcinoma"

_PeerJ, doi:10.7717/peerj.16197_

## Round 0.1 · original submission · Major Revisions

Authors should revise according to the suggestions of reviewers. The modifications should be marked. A point-to-point response letter is needed.

Reviewer 1 ·

Basic reporting

This manuscript presents a robust investigation of the long non-coding RNA (lncRNA) SNHG16 as a potential biomarker in head and neck squamous cell carcinoma (HNSCC). The authors innovatively focus on cuproptosis, a relatively new concept, thereby contributing significantly to the current understanding of lncRNAs in HNSCC. The application of a broad range of methodologies and the thorough contextualization of results within existing literature are commendable. The construction of a prognostic model and the identification of several cuproptosis-related lncRNAs, with potential roles in predicting HNSCC prognosis and immune status, further enhance the study's value.
While the study is commendable in its exploration of lncRNA SNHG16 in HNSCC, it would be strengthened by experimental validation of in silico predictions, a deeper understanding of the molecular mechanisms linking identified lncRNAs and cuproptosis, independent validation of the prognostic model, a thorough exploration of therapeutic implications, and clarification of the sample size to ensure reliable and applicable results.
Despite these limitations, the study offers important insights and opens up avenues for future research in the field of HNSCC. With appropriate modifications and further research, the authors' findings could significantly impact the diagnosis and treatment of HNSCC.

Experimental design

no comment

Validity of the findings

no comment

Additional comments

The study's flow chart does not offer crucial information and could be more suitably placed in the supplementary data.
Line 14 could be restructured for clarity. It could be read as "We aim to investigate the potential value of Cuprotosis-related lncRNA signaling in predicting clinical prognosis and immunotherapy, as well as its relationship with drug sensitivity in HNSCC."
Line 17, "We firstly identified" should be "We first identified."
Line 20, it would be clearer to state "We used RT-qPCR to validate our findings in a laryngeal squamous cell carcinoma cell line and in 12 pairs of laryngeal squamous cell carcinoma and adjacent normal tissues."
Line 25, it's not clear what "were suppressed" refers to. It may be better to specify, e.g., "which were found to be suppressed in patients with oncogene mutations in the high-risk group."
Line 26, it would be clearer to state "Patients with a high tumor mutation burden (TMB) exhibited poorer overall survival (OS)."
Line 34, "and immunotherapy" seems to hang at the end of the sentence and might need more context.
Line 40, the comma after "larynx" is unnecessary. It should read: "larynx and is strongly associated with smoking, alcohol consumption, and HPV infection."
Line 43, "has" should be "have" to match the plurality of the subject "lack of early detection and asymptomatic nature."
Line 85 could be restructured for clarity. It could be read as "RNA sequencing data, clinical data, and gene mutation data of 502 HNSCC and 44 normal head and neck tissue specimens were obtained from the Genomic Data Commons portal (https://portal.gdc.cancer.gov/)."
Line 87, "Then analyzed" is a fragment. It could be revised to "We then analyzed..."
Line 93, "is constructed" should be "was constructed"
Line 109, there is an error in "newly patients." It should be "new patients" or "patients who had recently undergone surgical treatment."
Line 114, "for patients who have" should be "for patients who had" to match the past tense used elsewhere in the text.
Line 116, "Two" should not be capitalized.
Line 119, "80. C" should be "80°C" for proper notation of temperature in Celsius.
Line 121, "paracancerous" is not a common term. Consider using "peri-tumoral" or "adjacent non-cancerous" instead.
The references should be update in time.
Line 123, "are evaluated" should be "were evaluated" to match the past tense used elsewhere in the text.
Line 132 could be restructured for better readability. It could be read as "Figure 1 depicts the workflow for establishing the prognostic model and subsequent analysis. Our study conducted a comprehensive investigation into the pivotal roles and prognostic implications..."
Lines 134-135 could be restructured for better readability. It could be read as "Initially, we collected mRNA and lncRNA expression profiles, along with clinical information from 502 HNSCC tissue samples and 44 non-tumor samples."
Line 137, "we visually represented the coexpression..." could be restructured for clarity. It could be read as "We then visually represented the coexpression..."
Lines 146-147 could be restructured for better readability. It could be read as "Figure 2E shows a correlation heatmap that provided additional evidence of the association between Cuproptosis-related genes and the lncRNAs that were incorporated into the prognostic model."
Lines 157-159 could be restructured for better readability. It could be read as "Figure 2K shows the heatmap of the level of 11 lncRNAs in the training group. For example, SNHG16, AC012184.3, LINC01012, AC243965.2, AP000866.1 are high-risk lncRNAs, while ECE1-AS1, AL358777.3, AL033527.2, AL162458.1, AC090587.1, CDKN2A-DT are low-risk lncRNAs."
Line 164 could be restructured for better readability. It could be read as "The results of the testing group suggest the reliability of the prognostic model we constructed."
Lines 167-168 could be restructured for better readability. It could be read as "Figure 3E shows that the univariate Cox regression analysis revealed a significant association between OS and age, histological grade, pathological stage, and risk score of HNSCC patients (P<0.05)."
Line 176 could be restructured for better readability. It could be read as "...signifying that our model was superior in forecasting the prognosis of HNSCC compared to the conventional clinicopathological features."
Lines 189-190 could be restructured for better readability. It could be read as "The enrichment analysis revealed significant Gene Ontology (GO) and Kyoto Encyclopedia of Genes and Genomes (KEGG) pathways for Cuproptosis-related lncRNAs."
Line 195 could be restructured for better readability. It could be read as "The KEGG analysis showed enrichment in the cytokine-cytokine receptor interaction, cell adhesion molecules, B cell receptor signaling pathway, and NF-kappa B signaling pathway."
Line 198 could be restructured for better readability. It could be read as "Figure 4GHI indicates that Cuproptosis-related lncRNAs are associated with immune cell infiltration in HNSCC."
Lines 201-203 could be restructured for better readability. It could be read as "Figure 5AB shows that the mutation frequency was generally higher in the high-risk group compared to the low-risk group (e.g. TP53: high-risk, 76%; low-risk, 57%. FAT1: high-risk, 24%; low-risk, 16%. CSMD3: high-risk, 22%; low-risk, 16%)."
Lines 206-207 could be restructured for better readability. It could be read as "Figure 5E shows that high tumor mutation burden (TMB) is associated with lower overall survival in patients."
Line 213 could be restructured for better readability. It could be read as "Figure 6A-C indicates that three drugs (PD-0332991, Cytarabine, and WH-4-023) had lower IC50s in the high-risk group, implying higher sensitivity to drug treatment."
Lines 218-220 could be restructured for better readability. It could be read as "Figure 7 shows the Kaplan-Meier analysis. It indicates that ECE1-AS1, AL033527.2, AL162458.1, and AC090587.1 were positively correlated with overall survival (OS), suggesting a good prognosis, while SNHG16 and AC012184.3 were negatively correlated with OS, indicating a poor prognosis."
Line 224 could be restructured for better readability. It could be read as "For experimental validation, we ultimately screened three prognostic lncRNAs with consistent expression.
Line 233 could be restructured for better readability. It could be read as "Conventional treatments can manage this prevalent malignant tumor to enhance patients' survival time and quality of life."
Lines 239-240 could be restructured for better readability. It could be read as "Research has indicated that lncRNAs play a role in various cancers, and different lncRNA predictive signatures can be used to predict the prognosis of cancer patients [15, 16]."
Lines 243-244 could be restructured for better readability. It could be read as "Our study identified several Cuprotosis-related lncRNAs, constructed a reliable prognostic model, and underlined their role in forecasting the prognosis and immune status of HNSCC patients."
Lines 246-247 could be restructured for better readability. It could be read as "We constructed a prognostic model by analyzing 11 prognostic lncRNAs through a series of biological analyses."
Lines 257-259 could be restructured for better readability. It could be read as "Small nucleolar RNA host gene 16 (SNHG16), another lncRNA, is expressed by the SNHG16 gene located on chromosome 17. It was first identified in human neuroblastoma tissue where it was found to promote malignant development [21, 22]."
Line 271 could be restructured for better readability. It could be read as "Our aim was to explore the role of these lncRNAs in HNSCC and their prognostic value."
Line 276 could be restructured for better readability. It could be read as "The nomograms showed that the model was a robust predictor of survival outcomes in HNSCC patients."
Line 282 could be restructured for better readability. It could be read as "Cytokines, including interleukins, interferons, tumor necrosis factor (TNF) superfamily members, chemokines, and growth factors, are potent secretory regulators involved in cell-cell communication during homeostasis and disease [26]."
Line 299 could be restructured for better readability. It could be read as "The NF-κB pathway is a receptor pathway regulated by proteolytic enzymes."
Lines 302-304 could be restructured for better readability. It could be read as "Activation of NF-κB has been observed in a variety of tumors, such as liver cancer, T-lymphocytic leukemia, breast cancer, cervical squamous cell carcinoma, ovarian cancer, thyroid cancer, Hodgkin’s lymphoma, small cell lung cancer, pancreatic cancer, and malignant melanoma [41]."
Lines 308-310 could be restructured for better readability. It could be read as "Immunotherapy is a promising approach for treating head and neck tumors. To guide this approach, we compared the expression of immune checkpoints between high- and low-risk groups. We observed significant differences in immune cells, immune checkpoint genes, and immune-related pathways."
Lines 313-314 could be restructured for better readability. It could be read as "We found that a high mutational burden coupled with high-risk scores was associated with a worse prognosis. Additionally, we observed that mutant TP53 expression promoted malignant transformation in the high-risk group [43, 44]."
Lines 317-318 could be restructured for better readability. It could be read as "Our findings suggest that high-risk patients may benefit from certain drugs such as PD-0332991, Cytarabine, and WH-4-023, while showing resistance to others such as YM201636, PIK-93, KIN001-102, Phenomin, and THZ-2-495. These findings lay the groundwork for precise and individualized treatment of head and neck tumors."
Lines 320-321 could be restructured for better readability. It could be read as "In summary, our study provides novel insights into the pathogenesis and progression of head and neck tumors, with a focus on copper homeostasis."
Lines 322-323 could be restructured for better readability. It could be read as "We identified cuproptosis biomarkers that could act as prognostic indicators for head and neck tumors and potentially assist in clinical decision-making."
Line 325 could be restructured for better readability. It could be read as "Cuproptosis is a recently discovered cell death mechanism, and there is limited research on related lncRNAs."

Reviewer 2 ·

Basic reporting

1 The authors should improve the language, especially grammer.
3 The review of related work is not sufficiently thorough and not sufficiently specific.
3 There are missing plenty of references in the manuscript.
4 There are missing details in 2. Method section, especially 2.1 and 2.2.

Experimental design

1 Many results are descriptive, and lacking quantitative information.
2 The authors should provide the p-value and sample size for each analysis, especially figure 8.

Validity of the findings

1 In today's research, open science is one of the essential aspects, and sharing the datasets and scripts/source codes is an important part. This way, future researchers can replicate your results when and where necessary. The authors should provide this information.
2 The authors should provide their important results in tables in addition to figures.

Additional comments

1 Discussion section should be improved to better reflect the quality of the work.

Reviewer 3 ·

Basic reporting

The manuscript by Han and coworkers titled ‘Cuproptosis-related LncRNA SNHG16 as a biomarker for diagnosis and prognosis of head and neck squamous cell carcinoma’ describes prognostic marker derived from long noncoding RNAs showing correlation of transcript expression with cuproptosis genes. The introduction and discussion are written clearly. The methods description and results section need extensive improvement. There are major issues with study design. The prognosis signature is the key portion of the manuscript, and this portion is poorly designed.
1. Methods description for prognosis signature development is very brief and vague. Please elaborate.
2. Similarly in the results section, the explanation prognosis signature discovery is vague. Please explain the results shown in figure 2B to 2E in detail.
3. What is the value of lambda selected from lasso cross validation? Is it lambda.min or lambda.1se?
4. The prognostic signature was discovered in TCGA dataset. But there is no validation for this in an independent dataset. It is a major drawback.

Experimental design

The major problem with the study is lack of validation. Successful validation of prognostic model in independent cohort is important to prove that the identified prognostic factors are reliable.

Validity of the findings

Even though the authors presented calibration curves which are very poor. These calibration curves suggest that the models are unreliable and poorly calibrated.

---

## Round 0.2 · Minor Revisions

Authors should revise according to the suggestions of reviewers. The modifications should be marked. A point to point response letter is needed.

Reviewer 1 has suggested that you cite specific references. You are welcome to add it/them if you believe they are relevant. However, you are not required to include these citations, and if you do not include them, this will not influence my decision.

Reviewer 1 ·

Basic reporting

I appreciate the authors' reply. Most of my comments and concerns have been successfully addressed. However, there are still a few that need to be addressed before considering publication.
The font size and type should be consistent in all figures. The font size of some figures is too small to read, and it would be better if the authors could enlarge them. The font type is usually Times New Roman or Arial.
In Figure 2 and 3, some panels that represent the internal analysis processes should be removed or moved to the supplementary data. Some of the meaningful panels should be enlarged and made clearer.
The figure legends are too simple and most of the information is lost. It would be great if the authors could enrich them.
The literature should be kept up-to-date, and it would be great if the authors updated their references to include more recent studies, such as PMID: 37081965, 37063265, doi.org/10.1016/j.clicom.2022.08.002, 35563442
The statistical significance labels in the figures should be consistent. Some figures are labeled with numerical values, while others are labeled with star symbols.
In Figures 8D, E, and F, do the data in the left and right panels come from the same detection or different detections? The left panel only shows tumor tissues. Where are the normal tissues?

Experimental design

None

Validity of the findings

None

Additional comments

None

Reviewer 2 ·

Basic reporting

My previous concerns have been addressed.

Experimental design

My previous concerns have been addressed.

Validity of the findings

My previous concerns have been addressed.

Additional comments

My previous concerns have been addressed.

---

## Round 0.3 · accepted · Accept

The authors have addressed the reviewers' concerns properly and revised the manuscript accordingly. The manuscript can be accepted for publication in its current form.